# Effect of Enzyme Modified Soymilk on Rennet Induced Gelation of Skim Milk

**DOI:** 10.3390/molecules23123084

**Published:** 2018-11-26

**Authors:** Kaixin Li, Jianjun Yang, Qigen Tong, Wei Zhang, Fang Wang

**Affiliations:** 1Beijing Laboratory for Food Quality and Safety, Food Science and Engineering College, Beijing University of Agriculture, Beijing 102206, China; joycelee_1108@163.com (K.L.); 18811626398@163.com (J.Y.); tongqigen@163.com (Q.T.); zhangwei300109@sina.com (W.Z.); 2Beijing Key Laboratory of Agricultural Product Detection and Control for Spoilage Organisms and Pesticides, Food Science and Engineering College, Beijing University of Agriculture, Beijing 102206, China

**Keywords:** soymilk, enzyme modified soymilk, skim milk, rennet induced gelation, cheese, rheological properties

## Abstract

In this study, soymilk was hydrolyzed to different degrees with flavourzyme, and then soymilk and enzyme modified soymilk at various levels were added to skim milk respectively, to generate a mixed gel using rennet. Rheological properties, scanning electron microscopy imaging, and physical and chemical indexes were examined to reveal the effect of enzyme modified soymilk on rennet induced gelation of skim milk. Results showed that soymilk inhibited the aggregation of skim milk, led to a decrease in storage modulus (G’), significantly increased moisture content and curd yield, and the resulting network was coarse. Enzyme modified soymilk with a molecular weight below 20 kDa led to a more uniform curd distribution, which counteracted the reduction of G’ and allowed for the formation of a stronger gel. Both the moisture content and the curd yield increased with the addition of soymilk and enzyme modified soymilk, and overall the effect of adding a high degree of hydrolysis of enzyme modified soymilk was superior. Compared to untreated soymilk, the addition of a certain amount of enzyme modified soymilk resulted in a new protein structure, which would improve the texture of blend cheese.

## 1. Introduction

In recent years, with rising food price levels, increased dietary awareness and health concerns, people are becoming more interested in soy products and novel value-added food products. Consuming foods that contain casein and soy protein has many beneficial health effects, and gels made from these two proteins can achieve dual health effects of two products [1].

The gelation of milk often occurs in two ways. In the first mechanism, during acidification, colloidal calcium phosphate is dissolved, and caseins formed a self-supporting network near their isoelectric point [2]. In the other mechanism, rennet specifically hydrolyzes ĸ-casein at the Phe105–Met106 bond, decreasing steric and electrostatic stabilization and resulting in casein aggregation [3,4]. Soymilk gels are typically prepared by heating soymilk, and gelation can also be induced by adding magnesium chloride or glucono-δ-lactone [5].

Many studies have focused on mixed skim milk and soymilk gels. Previous studies have shown that under acidification conditions, the viscoelastic properties and microstructure of generated protein gels of skim milk and soymilk mixtures are dependent on the concentration of skim milk powder and soy protein concentrate [6]. In addition, adding rennet to skim milk and soymilk systems, rennet will only hydrolyze the cow milk components [7]. It has been suggested that combined acid and rennet induced gelation of cow milk and soymilk mixtures would lead to the simultaneous aggregation of proteins [8].

Soy protein hydrolysate has been used in specialized adult nutrition formulations [9]. Many studies have shown when soy protein isolate was enzymed, its functional properties can be improved [10,11,12,13,14]. Rinaldoni et al. [15] added soy protein to skimmed cow milk to develop a spreadable cheese-like product. Their results showed that soy protein concentrate improved cheese yield, but the elastic properties were significantly reduced. Gao et al. [16] studied the enzymatic hydrolysis process of soymilk, and mixed enzyme modified soymilk with cow milk to make mozzarella cheese. It was found that the stretchability, elasticity and hardness of mozzarella cheese had been greatly improved. For most cheeses, rennet induced gelation is the key process affecting cheese yield and quality [17,18,19]. Therefore, the objective of this study was to explore the effect of different amounts and hydrolysis degrees of enzyme modified soymilk on renneting of skim milk, to provide a theoretical basis for the production of a new blend cheese.

## 2. Results

### 2.1. Protein Structural Characteristics

The effect of enzymatic hydrolysis on protein profiles of skim milk, soymilk and enzyme modified soymilk is shown in Figure 1, and lanes 1–5 are the result of the marker, skim milk, soymilk, and high and low degree of hydrolysis of enzyme modified soymilk, respectively. Compared to soymilk, the distribution of bands of enzyme modified soymilk changed dramatically, indicating the decomposition of soy protein. After enzymatic hydrolysis of soy protein, the bands were mostly concentrated below 20 kDa, and its β-conglycinin, and acidic and basic subunits of glycinin almost disappeared entirely. Compared to the low degree of hydrolysis of enzyme modified soymilk, the molecular weight of soy protein further reduced after the high degree of hydrolysis.

### 2.2. Rheological Properties

Rheological properties were used to monitor the effect of soymilk and enzyme modified soymilk on the coagulation process of skim milk. After adding rennet, the storage modulus of the control increased rapidly, and then changed slowly with gelation time (Figure 2a–c). The storage modulus of experimental samples (except S2 and S3) showed the same trend. However, the storage modulus of experimental samples was lower than that of the control sample at the same renneting time (*p* < 0.05). The trend of the changes of loss modulus was consistent with that of storage modulus (data not shown). The loss tangent (the ration of loss modulus to storage modulus) decreased with renneting (Figure 2d–f). Compared to the control, gelation time of experimental samples lagged significantly (*p* < 0.05) (Table 1), and the delay effect in gelation time was more significant (*p* < 0.05) with the increase in soymilk and enzyme modified soymilk. The order of gelation time of experimental samples at the same concentration was L group ≥ H group > S group. At the end of gel formation, compared with control sample, the final strength of the curd determined by the storage modulus value at 60 min (G’_60min_) in experimental samples decreased significantly (*p* < 0.05) (Table 1), and the decrease was more significant with the increase in soymilk and enzyme modified soymilk (*p* < 0.05). The order of G’_60min_ of the experimental samples at the same concentration was L group > H group > S group.

### 2.3. Physical and Chemical Indicators

The addition of soymilk and enzyme modified soymilk significantly increased the moisture content and curd yield (Table 2). With the increase in soymilk, curd yield showed a significant increase (*p* < 0.05), but the moisture content did not change significantly (*p* > 0.05). With the increase in enzyme modified soymilk, the moisture content and curd yield increased significantly (*p* < 0.05). At 5% and 10% addition, the moisture content and curd yield of the S group were significantly higher than those of L group and H group (*p* < 0.05). However, no significant differences were found in moisture content and curd yield among the three groups at 15% addition (*p* > 0.05), and moisture content and curd yield showed no significant differences between L group and H group at 20% addition (*p* > 0.05). Under 25% addition, the curd yield was higher in L group than in H group, and moisture content between the two samples showed no significant difference (*p* > 0.05).

### 2.4. Microstructure

The above results showed that rennet induced gelation was affected by soymilk and enzyme modified soymilk. Therefore, the ratio of skim milk and soymilk/enzyme modified soymilk selected here was 85:15, to ensure their comparability. The caseins in the control were relatively smooth (Figure 3a). The soy protein in the sample with soymilk attached to the surface of caseins, formed rough and small clusters, and therefore resulted in a coarse structure (Figure 3b). Compared with the control and S3, the addition of enzyme modified soymilk caused a more uniform protein network (Figure 3c,d). Compared to the control, S3 and L3, the casein aggregates in H3 were smaller, and the small voids became denser. The soy protein in S3 and enzyme modified soy protein in L3 mainly existed in large caseins, but the distribution of enzyme modified soy protein in H3 was more uniform.

## 3. Discussion

In our study, the pH values during renneting among mixtures showed no significant differences (data no shown), and the gel formation was the action of rennet. Under the conditions of our study, the addition of rennet to soymilk did not result in gelation, as monitored using a rheometer with no change in storage modulus (data not shown), which was agreed with previous studies [9,20]. The increase in the storage modulus and the decrease in loss tangent indicated that there was an interaction among caseins, and the increased contact area of caseins therefrom led to the formation of a gel network [21]. Previous studies have shown that rheological properties were mainly affected by casein concentration [22]. In experimental groups, the decrease in skim milk, which was partly replaced with soymilk or enzyme modified soymilk, resulted in the decrease in ĸ-casein and therefore the number of action sites for rennet. Moreover, soymilk and enzyme modified soymilk set a barrier to the accumulation of caseins and further affected their aggregation (Figure 4). Thus, we speculated that these two effects would lead to a lag in gelation time and a decrease in final strength, which was also reported in previous literature [23,24]. Meanwhile, the delay in gelation time was more significant and final strength further decreased with the increase of soymilk and enzyme modified soymilk. Compared to the control with the final strength of 96.26 Pa, when soymilk content increased to 15%, the final strength was reduced to 3.55 Pa (Table 2), indicating that aggregation of caseins was strongly hindered [8]. However, this effect can be overcome to a certain extent by adding enzyme modified soymilk, showing that the final strength of L3 and H3 was 75.32 Pa and 65.84 Pa, respectively. Compared with the control and S1, the decrease of ĸ-casein in L1 and H1 had little effect on casein aggregation. The main contributor may be the reduction of protein size in enzyme modified soymilk as monitored by sodium dodecyl sulfate polyacrylamide gel electropheresis (SDS-PAGE), which was similar to the report of Luo et al. [25]. They studied the effect of native fat globule size on gel formation, and found storage modulus of the curd with small fat globules was higher than that of the curd with large fat globules. The high level of enzymatic hydrolysis increased the number of peptides, which may shield enzymatic sites and inhibit the aggregation of caseins (Figure 4), and therefore final strength in H group was significantly lower than that of L group.

Previous studies have shown that the addition of soy protein to milk can increase the moisture content of the resulting cheese [26], which was consistent with our result. Soy protein had water-holding properties [9], and therefore we speculated that soy protein trapped in the gel structure can retain more moisture through hydrogen bonds and other forces. Furthermore, our rheological results have shown that soymilk significantly inhibited rennet induced aggregation of caseins, affected dehydration of curds and increased moisture content and curd yield, which was in agreement with other studies [27,28,29]. Utsumi [30] indicated that the water-holding property of soy protein after enzymatic hydrolysis can be improved. However, the moisture content in L group and H group was lower than that in S group in our study, which may be ascribed to the decreased inhibition of gelation and the loss of small substances after enzymatic hydrolysis (Figure 4). As the amount of soymilk increased, the hydrophilicity of soy protein and the inhibition effect on casein aggregation were more significant, and therefore moisture content and curd yield further increased. There was no significant difference in moisture content between L group and H group under the same addition, but the curd yield of H group was lower than that of L group, and the losing of smaller peptide segments or amino acids with whey after a high degree of hydrolysis may be the major contributor.

The control curd showed a compact protein matrix, which was composed of thick chains and large clusters of caseins (Figure 3a), and it was consistent with previous report [31]. The soy protein adhered or was bound to the surface of caseins, formed a clustered structure and therefore inhibited the aggregation of caseins, causing the increased moisture content and curd yield. The research of Ingrassia et al. showed a similar phenomenon [32]. Both glycinin and β-conglycinin in soymilk can be effectively degraded by flavourzyme [33], which may decrease the inhibition of gelation and therefore contribute to a more compact and orderly gel structure. The increase in the number of small substances after a high degree of hydrolysis further inhibited the aggregation of caseins, and thus decreased the size of casein aggregates.

## 4. Materials and Methods

### 4.1. Skim Milk Preparation

Fresh milk was supplied by a local farm (Fuchun Farm, Beijing Sanyuan Food Co., Ltd., Beijing, China) and sodium azide (0.02%) was added immediately. The milk was centrifuged at 4000× *g* for 20 min at 4 °C (LYNX 2000, Thermo Scientific, Waltham, MA, USA) and then filtered three times through filters (Fisher Scientific, Whitby, ON, Canada), as far as possible to remove fat. The resulting skim milk was pasteurized at 63 °C for 30 min, cooled to room temperature, and stored in a refrigerator until use. The fat content of skim milk was 0.04 ± 0.01% by the AOAC method [34], and protein content was 3.21 ± 0.05% measured using the Kjeldahl method [19].

### 4.2. Enzyme Modified Soymilk Preparation

Soybeans (Helen soybean, Heilongjiang Black Soil Town Modern Agriculture Development Co., Ltd., Haerbin, Heilongjiang, China) containing 36% protein and 16% fat were obtained from a local market. Soymilk was prepared as previously reported [35], with slight modifications. Soybeans were soaked overnight in deionized water for hydration and mixed with a certain amount of deionized water to obtain the desired protein content. Subsequently, samples were passed through a household soymilk maker (RM-125, Ruimei, Wuxi, Jiangsu, China). After soymilk maker cycle was completed, soymilk was passed through a filter (Fisher Scientific, Whitby, ON, Canada) and passed through cheesecloth to remove okara. Then soymilk was boiled at 100 °C for 10 min and rapidly cooled to room temperature. The protein content of the obtained soymilk was 3.32 ± 0.07% measured using the Kjeldahl method [19].

Flavourzyme (0.28%; HP202474, Novozymes, Tianjin, China) was added to soymilk for 2 h and 4 h at 45 °C, respectively. The subsequent degree of hydrolysis of soymilk was 5.92% and 9.88% according to the Ninhydrin method [36]. L and H were used to represent the low (L) and high (H) degree of hydrolysis. The enzyme modified soymilk was then incubated at 85 °C for 15 min to inactivate the enzyme. When samples were cooled to room temperature, sodium azide (0.02%) was added. The obtained samples were stored at 4 °C until use.

### 4.3. Electrophoresis

The effect of enzyme treatments on protein profiles of soymilk and enzyme modified soymilk was determined using SDS-PAGE with separating and stacking gels containing 15 and 4% acrylamide, respectively. A molecular weight marker ranging from 11 to 180 kDa (PR1910, Solarbio, Beijing, China) was used as a standard. Skim milk, soymilk and enzyme modified soymilk were dissolved with sample buffer (10 mM DTT, pH 6.8, 1 mM EDTA, 1% SDS, 10% glycerol, and 0.01% bromphenol blue) and then boiled for 5 min. After centrifugation, the electrophoresis was carried out following the method of Lamsal et al. [37].

### 4.4. Mixture Preparation

The mixtures were prepared according to Table 3, and skim milk was selected as the control. For convenient reading, samples with different ratios of skim milk and soymilk/enzyme modified soymilk are indicated by **S1**–**S3**, **L1**–**L5** and **H1**–**H5**. The mass ratio selected were to ensure that rennet induced gelation would occur. After the mixtures of skim milk and soymilk/enzyme modified soymilk had been prepared, the samples were stirred at room temperature for 30 min, and the pH was adjusted to 6.7 with 0.1 M NaOH. All the samples were stored in a refrigerator until use.

### 4.5. Curd Making

All samples (weighed and recorded as W_1_) were first warmed at 32 °C for 30 min and then incubated with rennet (0.01%). Rennet (CHY-MAX powder Extra) was got from Chr. Hansen (Beijing, China) and the coagulation strength was ~2235 IMCU/g. After 1 h of incubation, the resulting gels were cut manually into small pieces (1 × 1 × 1 cm), and then set for 5 min to promote syneresis. Curds were subsequently centrifuged at 4000× *g* for 15 min at room temperature. The upper whey was removed and the curd was collected carefully and weighed (recorded as W_2_). The curd yield was calculated using following Equation:Curd yield = W_2_/W_1_ × 100%

The moisture content of the curds was analyzed using an oven method [34]. All measurements were obtained in triplicate.

### 4.6. Rheological Properties

The coagulation process of the mixtures was monitored as described previously [38]. All samples were first maintained at 32 °C for 30 min. The gelation time was defined as the time point when storage modulus value of the gel was ≥1 Pa [6]. All measurements were determined in triplicate.

### 4.7. Microstructure Determination

The samples for microstructure determination were prepared as described earlier [39]. Then samples were critical point dried in a Technics Critical Point Dryer (CPD030, Leica Mikrosysteme GmbH, Wien, Austria), fixed on the sample table (MC1000, Hitachi, Ibarakiken, Japan) for spraying operation and then examined using a scanning electron microscope (SU8010, Hitachi, Ibarakiken, Japan) operated at 25 kV. Representative micrographs were selected for presentation.

### 4.8. Statistical Analysis

Statistical analysis of the data was performed using SPSS 18.0 (SPSS Inc., Armonk, NY, USA). The differences were compared at a significance level of *p* < 0.05.

## 5. Conclusions

Soymilk and enzyme modified soymilk have a significant effect on renneting of skim milk. Compared to skim milk, a certain proportion of soymilk and enzyme modified soymilk would decrease storage modulus of the mixed gel, and significantly increase moisture content and curd yield. In addition, with the increase in soymilk and enzyme modified soymilk, the inhibition of gel formation was stronger, and moisture content and curd yield further increased. Compared to soymilk, enzyme modified soymilk with more small molecular substances decreased the impediment of rennet induced gelation, improved storage modulus of the gel, and the resulting curd had a more uniform structure.

## Figures and Tables

**Figure 1 molecules-23-03084-f001:**
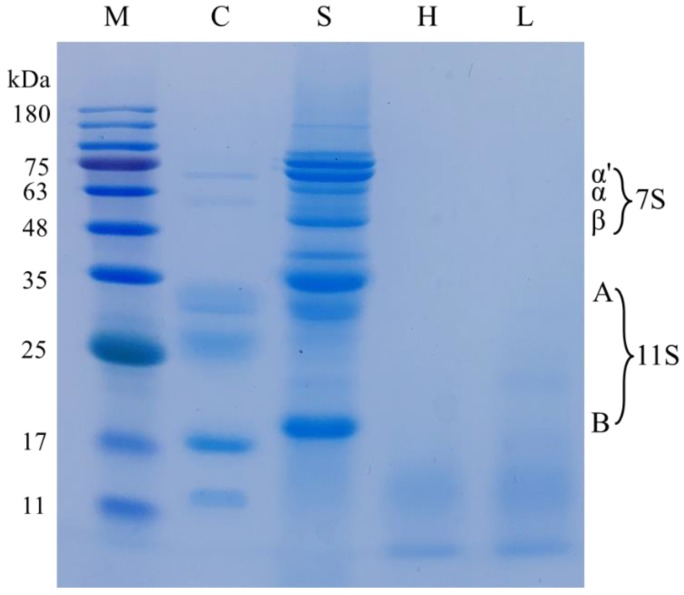
Sodium dodecyl sulfate polyacrylamide gel electropheresis for the control, soymilk and enzyme modified soymilk. M, marker; C, skim milk; S, soymilk; H, high degree of hydrolysis of soymilk; L, low degree of hydrolysis of soymilk. 7S and 11S indicate the relative proteins in soymilk.

**Figure 2 molecules-23-03084-f002:**
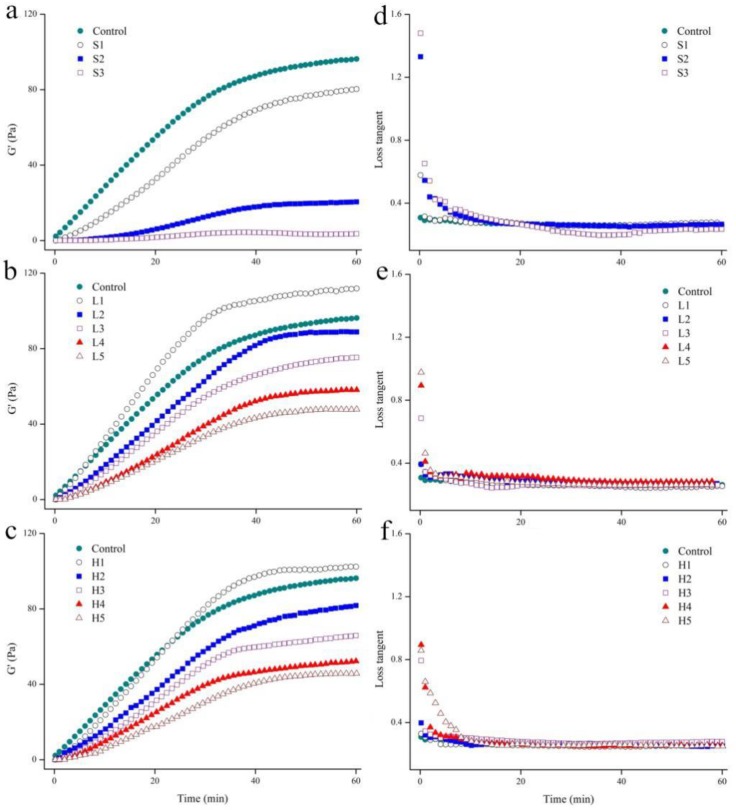
The changes in storage modulus (G’) (**a**–**c**) and loss tangent (**d**–**f**) during rennet induced gelation. Curves are representative runs.

**Figure 3 molecules-23-03084-f003:**
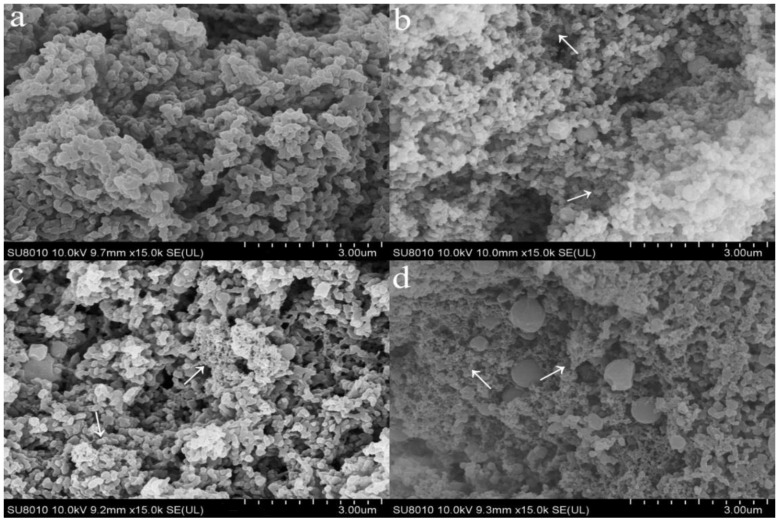
Scanning electron microscopy micrographs of rennet induced curds. **a**, control; **b**, S3; **c**, L3; **d**, H3 (magnification: ×15,000) (The arrow denotes soy protein and its hydrolysate).

**Figure 4 molecules-23-03084-f004:**
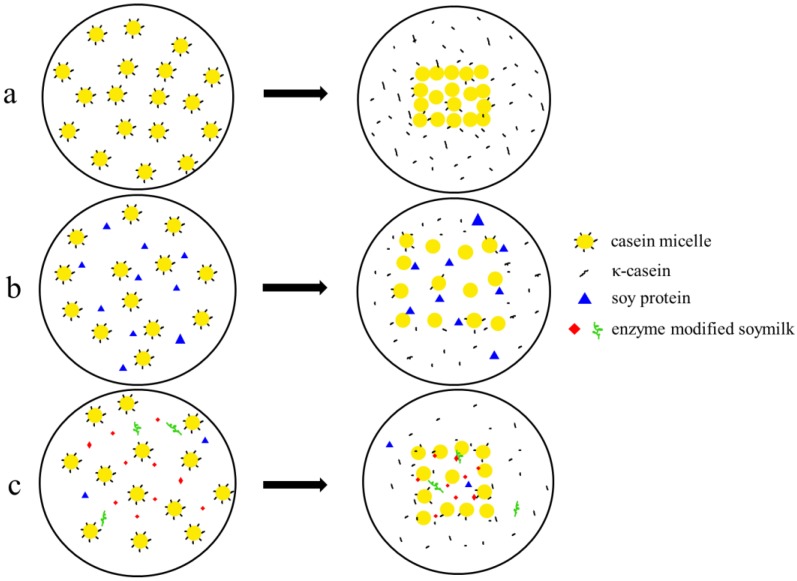
Schematic diagram of rennet induced gelation of different mixtures. **a**, skim milk; **b**, skim milk and soymilk; **c**, skim milk and enzyme modified soymilk.

**Table 1 molecules-23-03084-t001:** The rheological parameters of mixtures during renneting.

Sample Number	G’_60min_ ^1^ (Pa)	Gelation Time ^2^ (min)
**Control**	96.26 ± 1.56 ^k^	0.36 ± 0.13 ^a^
**S1**	80.33 ± 1.86 ^i^	1.07 ± 0.16 ^c^
**S2**	20.49 ± 1.40 ^b^	8.14 ± 0.15 ^i^
**S3**	3.53 ± 0.95 ^a^	16.28 ± 0.11 ^j^
**L1**	111.85 ± 1.68 ^m^	0.78 ± 0.08 ^b^
**L2**	88.86 ± 1.77 ^j^	1.02 ± 0.20 ^bc^
**L3**	75.32 ± 1.33 ^h^	1.53 ± 0.13 ^d^
**L4**	58.14 ± 1.94 ^f^	2.28 ± 0.14 ^e^
**L5**	47.88 ± 1.40 ^d^	3.04 ± 0.17 ^g^
**H1**	102.47 ± 1.46 ^l^	0.83 ± 0.09 ^bc^
**H2**	81.74 ± 1.24 ^i^	1.07 ± 0.13 ^c^
**H3**	65.84 ± 0.88 ^g^	1.72 ± 0.13 ^d^
**H4**	52.31 ± 1.78 ^e^	2.68 ± 0.11 ^f^
**H5**	45.14 ± 1.18 ^c^	3.55 ± 0.18 ^h^

Means with different letters within the same column are significantly different (*p* < 0.05). ^1^ G’_60min_ means storage modulus value of the gel at the end of renneting at 60 min. ^2^ Gelation time means the time point when storage modulus value of the gel was ≥1 Pa.

**Table 2 molecules-23-03084-t002:** The moisture content and curd yield of rennet induced curds.

Sample Number	Moisture Content (%)	Curd Yield (%)
**Control**	75.25 ± 3.17 ^a^	13.52 ± 0.32 ^a^
**S1**	78.78 ± 0.78 ^cd^	15.20 ± 0.20 ^de^
**S2**	78.70 ± 0.51 ^cd^	15.71 ± 0.12 ^f^
**S3**	78.01 ± 0.65 ^bcd^	15.40 ± 0.15 ^ef^
**L1**	75.16 ± 1.37 ^a^	14.34 ± 0.17 ^bc^
**L2**	75.90 ± 1.74 ^ab^	14.99 ± 0.34 ^d^
**L3**	76.42 ± 0.84 ^abc^	15.61 ± 0.18 ^f^
**L4**	77.24 ± 1.40 ^abcd^	16.50 ± 0.18 ^g^
**L5**	78.74 ± 0.94 ^cd^	17.06 ± 0.28 ^h^
**H1**	76.89 ± 0.19 ^abc^	14.05 ± 0.12 ^b^
**H2**	75.76 ± 0.31 ^ab^	14.58 ± 0.11 ^c^
**H3**	77.52 ± 0.49 ^abcd^	15.51 ± 0.21 ^ef^
**H4**	77.68 ± 1.49 ^abcd^	16.41 ± 0.22 ^g^
**H5**	79.87 ± 2.29 ^d^	16.34 ± 0.25 ^g^

Means with different letters within the same column are significantly different (*p* < 0.05).

**Table 3 molecules-23-03084-t003:** The mixtures with different mass ratios of skim milk and soymilk/enzyme modified soymilk.

Mass Ratio	Soymilk	Low Degree of Hydrolysis of Soymilk	High Degree of Hydrolysis of Soymilk
95:5	**S1**	**L1**	**H1**
90:10	**S2**	**L2**	**H2**
85:15	**S3**	**L3**	**H3**
80:20	**-**	**L4**	**H4**
75:25	**-**	**L5**	**H5**

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
