# Peer review of "Effect of Enzyme Modified Soymilk on Rennet Induced Gelation of Skim Milk"

_molecules, 2018, doi:10.3390/molecules23123084_

Round 1

Reviewer 1 Report

The manuscript entitled “Effect of enzyme modified soymilk on rennet induced gelation of skim milk” aimed to evaluate the rheological characters, scanning electron microscopy imaging, and physical and chemical indexes to reveal the effect of enzyme modified soymilk on rennet induced gelation of skim milk

In fact, the current study conducted well but needs several corrections (Major revision) to be more acceptable for publication and my suggested corrections are:

1.     The similarity factor is extremely high about 24% decrease it to become less than 15%.

2.     In page 1 line 93 you mentioned (and cause them aggregation) rephrase this part.

3.     In page 2 line 47, you wrote (In addition, adding rennet to skim milk and soymilk systems, rennet will only hydrolyze the dairy components) rephrase this sentence.

4.     Remove the paragraph in page 2 which started at line 65 to the discussion part.

(The kinetics of rennet induced coagulation were revealed by determining rheological properties of skim milk and soymilk mixtures. The microstructure of the hybrid curds was observed using scanning electron microscope, and physical and chemical indexes of the mixed curds were also measured).

5.     In page 3 the authors mentioned “The soybeans containing 36% protein and 16% fat were purchased from a local grocery store” Kindly add the name of the manufacturer and the country for product

6.     Shift Table 3 to the end of the 4.4. Mixture preparation

7.     References number 9-14, 20, 22, 26 are too old if possible change them to more updated references.

8.     Major language and editing corrections required and to be checked by native speaker or any proof editing service to be this manuscript looked ideal.

9.     Other required corrections conducted by reviewer in the attached manuscript.

Author Response

Point 1: The similarity factor is extremely high about 24% decrease it to become less than 15%.

Response 1: We are appreciated for your valuable advice. We have revised our paper, and ensured that the similarity was less than 18%.

Point 2: In page 1 line 39 you mentioned (and cause them aggregation) rephrase this part.

Response 2: We are appreciated for your valuable advice. Considering the reviewers suggestion, we have deleted this part.

Point 3: In page 2 line 47, you wrote (In addition, adding rennet to skim milk and soymilk systems, rennet will only hydrolyze the dairy components) rephrase this sentence.

Response 3: We are appreciated for your valuable advice, and we have made correction in lines 45-46.

Point 4: Remove the paragraph in page 2 which started at line 65 to the discussion part.

(The kinetics of rennet induced coagulation were revealed by determining rheological properties of skim milk and soymilk mixtures. The microstructure of the hybrid curds was observed using scanning electron microscope, and physical and chemical indexes of the mixed curds were also measured).

Response 4: We are appreciated for your valuable advice. Considering the reviewers suggestion, we have deleted this part.

Point 5: In page 3 the authors mentioned “The soybeans containing 36% protein and 16% fat were purchased from a local grocery store” Kindly add the name of the manufacturer and the country for product.

Response 5: We are appreciated for your valuable advice. We have added the name of the manufacturer and the country of soybeans in lines 193-194.

Point 6: Shift Table 3 to the end of the 4.4. Mixture preparation

Response 6: We are appreciated for your valuable advice. Considering the reviewers suggestion, we have shifted Table 3 to the end of the 4.4. Mixture preparation.

Point 7: References number 9-14, 20, 22, 26 are too old if possible change them to more updated references.

Response 7: We are appreciated for your valuable advice. Considering the reviewers suggestion, we have changed some of the references to more updated references. The updated references were as follows.

[10] Ortiz, S.E.M.; Wagner, J.R. Hydrolysates of native and modified soy protein isolates: structural characteristics, solubility and foaming properties. Food Res. Int. 2001, 35, 511–518. doi: 10.1016/S0963-9969(01)00149-1

[11] Achouri, A.; Zhang, W. Effect of succinylation on the physicochemical properties of soy protein hydrolysate. Food Res. Int. 2001, 34, 507–514. doi: 10.1016/S0963-9969(01)00063-1

[12] Tsumura, K.; Saito, T.; Tsuge, K.; Ashida, N.; Kugimiya, W.; Inouye, K. Functional properties of soy protein hydrolysates obtained by selective proteolysis. LWT-Food Sci. Technol. 2005, 38, 255–261. doi: 10.1016/j.lwt.2004.06.007

[14] Surówka, K.; Zmudziński, D.; Surówka, J. Enzymic modification of extruded soy protein concentrates as a method of obtaining new functional food components. Trends Food Sci. Tech. 2004, 15, 153–160. doi: 10.1016/j.tifs.2003.09.013

[22] Ferrer, M.A.; Hill, A.R.; Corredig, M. Rheological properties of rennet gels containing milk protein concentrates. J. Dairy Sci. 2008, 91, 959–969. doi: 10.3168/jds.2007-0525

[26] Mahdy A.; Xia W.; Zhang G. Effect of soy protein supplementation on the quality of ripening Cheddar-type cheese. J. Dairy Sci. 2004, 57, 209–214. doi: 10.1111/j.1471-0307.2004.00107.x

Point 8: Major language and editing corrections required and to be checked by native speaker or any proof editing service to be this manuscript looked ideal.

Response 8: We are appreciated for your valuable advice. Considering the reviewers suggestion, we have re-edited the language of our paper in the English editing service of MDPI.

Reviewer 2 Report

molecules-388206

The manuscript molecules-388206 with the title of “Effect of enzyme modified soymilk on rennet induced gelation of skim milk” studied the influence of different degree enzyme modified soymilk addition on the gelation of skim milk. However, the changes of several parameters during gelation need to be provided in order to verify the gelation mechanism. The pH values during gelation should be added. The values of G” (loss modulus) and tan(delta) during gelation should also be added. Moreover, the concentration of skim milk was the main factor affected the rheological properties of curd. The comparison of these properties should be based on the same concentration of skim milk.

Detailed comments:

Keywords

P1L26 Add “cheese”.

P4 Figure 2 Add G” and tan(delta), and discuss the results in the text.

P7 Figure 3 Indicate the location of “casein protein”.

P9 Figure 4 These diagrams did not represent the gelation of solution with different mass ratio of skim milk and soymilk. Please revise.

P10L190, L210 According to the regulation, the preservative should not be added into fresh milk and soymilk.

P10L195 Provide the fat content of skim milk.

Author Response

Point 1: The pH values during gelation should be added. Moreover, the concentration of skim milk was the main factor affected the rheological properties of curd. The comparison of these properties should be based on the same concentration of skim milk.

Response 1: In fact, we have monitored the change of pH values during gelation. Considering the reviewers suggestion, we have added this statement in lines 129-130.

We are appreciated for your valuable advice. It was true as reviewer suggested that the concentration of skim milk was the main factor affected rheological properties of curds. Therefore, in our study, we have compared rheological properties of all samples, especially the samples on the same concentration of skim milk, such as in lines 87-88 and lines 92-93.

Point 2: P1L26 Add “cheese”.

Response 2: We are appreciated for your valuable advice, and we have made correction in line 26.

Point 3: Figure 2 Add G” and tan(delta), and discuss the results in the text.

Response 3: We are appreciated for your valuable advice. The trend of the changes in G'' was consistent with that of G', and therefore we didnt show it in our paper. Considering the reviewer’s suggestion, we have added this statement in lines 82-83.

Considering the reviewer’s suggestion, we have added the results and discussion of tan(delta) in Figure 2d-f, lines 83-84 and line 133.

Point 4: P7 Figure 3 Indicate the location of “casein protein”.

Response 4: We are appreciated for your valuable advice. In this experiment, we used soymilk and enzyme modified soymilk to instead part of skim milk. The main structure of the protein network were caseins which were shown in the following figure, and thus we only labeled soy protein and enzyme modified soy protein in our paper.

Point 5: P9 Figure 4 These diagrams did not represent the gelation of solution with different mass ratio of skim milk and soymilk. Please revise.

Response 5: We are appreciated for your valuable advice. In the mixtures with different ratio of skim milk and soymilk/enzyme modified soymilk, the numbers and ratios of proteins were different, however, the gelation process was more or less the same and it can be represented by the same diagram. The same representing method was also used in previous studies [1-5].

[1] Choi, J.; Horne, D.S.; Lucey, J.A. Effect of insoluble calcium concentration on endogenous syneresis rate in rennet-coagulated bovine milk. J. Dairy Sci. 2015, 98, 5955-5966.

[2] Anema, S.G.; Lee, S.K.; Klostermeyer, H. Rennet-induced aggregation of heated pH-adjusted skim milk. J. Agri. Food Chem. 2011, 59, 8413-8422.

[3] Wang, F.; Liu, X.; Hu, Y.; Luo, J.; Lv, X.; Guo, H.; Ren, F. Effect of carrageenan on the formation of rennet-induced casein micelle gels. Food Hydrocolloid. 2014, 36, 212-219.

[4] Sun, J.; Ren, F.; Chang, Y.; Wang, P.; Li, Y.; Zhang, H.; Luo, J. Formation and structural properties of acid-induced casein-agar double networks: role of gelation sequence. Food Hydrocolloid. 2018.

[5]  Li, H.; Yang, C.; Chen, C.; Ren, F.; Li, Y.; Mu, Z.; Wang, P. The use of trisodium citrate to improve the textural properties of acid-induced, transglutaminase-treated micellar casein gels. Molecules 2018, 23, 1632.

Point 6: P10L190, L210 According to the regulation, the preservative should not be added into fresh milk and soymilk.

Response 6: We are appreciated for your valuable advice. It was true as reviewers suggested that the preservative should not be added to milk in food production. But sodium azide can be added to fresh milk and soymilk to prevent microbial growth in the scientific research, which was previously reported in many studies [6-10].

[6] Kethireddipalli, P.; Hill, A.R.; Dalgleish, D.G. Protein interactions in heat-treated milk and effect on rennet coagulation. Int. Dairy J. 2010, 20, 838-843.

[7] Lin, Y.; Kelly, A.L.; OMahony, J.A.; Guinee, T.P. Addition of sodium caseinate to skim milk increases nonsedimentable casein and causes significant changes in rennet-induced gelation, heat stability, and ethanol stability. J. Dairy Sci. 2017, 100, 908-918.

[8] Roesch, R.; Juneja, M.; Monagle, C.; Corredig, M. Aggregation of soy/milk mixes during acidification. Food Reas. Int. 2004, 37, 209-215.

[9] Giroux, H.J.; Bouchard, C.; Britten, M. Combined effect of renneting pH, cooking temperature, and dry salting on the contraction kinetics of rennet-induced milk gels. Int. Dairy J. 2014, 35, 70-74.

[10] Koutina, G.; Knudsen, J.C.; Andersen, U.; Skibsted, L.H. Influence of colloidal calcium phosphate level on the microstructure and rheological properties of rennet-induced skim milk gels. LWT 2015, 63, 654-659.

Point 7: P10L195 Provide the fat content of skim milk.

Response 7: We are appreciated for your valuable advice, and we have added the fat content of skim milk in line 190

Round 2

Reviewer 1 Report

The authors did all the required corrections and I have not any further comments.

Reviewer 2 Report

The authors have been revised the manuscript according to the reviewer’s suggestion.